# Phytoremediation Effect and Soil Microbial Community Characteristics of Jiulong Iron Tailings Area, Jiangxi

Lingyu Hou [1,2,3], Wenzheng Wang [1,2,3], Liguo Song [1,2,3], Qian Wang [1,2,3], Xiangrong Liu [1,2,3], Yanlin Zhang [1,2,3] and Qiwu Sun [1,2,3,*]

1   Research Institute of Forestry, Chinese Academy of Forestry, Beijing 100091, China; houlingyu@caf.ac.cn (L.H.); 17600368768@163.com (L.S.)
2   Key Laboratory of Tree Breeding and Cultivation of State Forestry Administration, Chinese Academy of Forestry, Beijing 100091, China
3   State Key Laboratory of Efficient Production of Forest Resources, Chinese Academy of Forestry, Beijing 100091, China
*   Correspondence: sqw@caf.ac.cn

**Abstract:** The aim of this paper was to explore the remediation effect and mechanism of Wetland pine (*Pinus elliottii*), Chinese fir (*Cunninghamia lanceolata (Lamb.) Hook*), and Alder (*Alnus cremastogyne Burkill*) on heavy metal contaminated soil in the iron tailings of Jiulong Iron Tailings Area. At the same time, the specificity of plant rhizosphere and non-rhizosphere soil microbial community structure and ecological function were analyzed based on macrogenomic sequencing. The results showed that the dominant microbial genera in J1 (control) was Acidobacteria, followed by Proteobacteria and Actinobacteria. The microbial genera with the highest percentage of relative abundance in J2, J3, J4, J5, and J6 (J2, Wetland Pine rhizosphere; J3, Wetland Pine non-rhizosphere; J4, Chinese fir rhizosphere; J5, Alder rhizosphere; J6, Alder non-rhizosphere) were Proteobacteria, followed by Acidobacteria, and Actinobacteria. It was found that Proteobacteria promoted heavy metal solubilization, activated heavy metals, and converted their forms to improve plant uptake of heavy metals. This proves that the microorganisms of Proteobacteria are the key microbial genera in the study of regional heavy metal remediation. The antibiotic resistance genes (ARGs) in microorganisms can respond to the inducement of heavy metals. Here, we investigated the relationship between the abundance of soil microorganisms ARGs and heavy metal pollution in Jiulong Iron Tailings Area. There are significant differences in the quantity and category of ARGs in the rhizosphere and non-rhizosphere soil samples of the three tree species. The results of this study provide the foundation for the theory and practice of remediation of heavy metal contamination in soils of iron tailing areas in Jiulong Iron Tailings Area in similar stand conditions.

**Keywords:** heavy metal; metagenomics; phytoremediation effect; microbial community

## 1. Introduction

In recent years, more attention has been paid to environmental pollution. Heavy metal pollution is one of numerous environmental problems worldwide [1]. The long-term, high intensity, and large-scale development of mineral resources has met the resource needs of economic development to a certain extent and has played a great role in promoting the process of China's economic and social development. Although China is rich in iron ore reserves, the utilization rate of the ore is low due to the ore grade, production technology, and other reasons [2]. A large number of iron ore tailings have not been treated and utilized resourcefully on time, leading to a sharp increase in tailings stockpiles, which has caused a series of environmental problems, such as soil heavy metal pollution [3], water pollution [4], air pollution [5], etc. Therefore, it is important to find a suitable method to treat tailings and reduce their risk to the surrounding environment and residents.



A large number of studies have shown that there are four effective methods for tailings treatment: physical/chemical remediation, phytoremediation, microbial remediation, and plant microbial combined remediation [6–8]. Phytoremediation is a biological control and is a natural phenomenon in which plants are used to eliminate contaminants from the surroundings [9]. In recent years, the ability of microorganisms to reduce and absorb pollutants has been gradually found and confirmed. Compared with other methods, it requires low cost, repair speed, and will not cause secondary pollution. It is an effective means of heavy metal pollution treatment [10,11]. The heavy metal remediation mechanism of microorganisms mainly uses the complexation, adsorption, and redox between microorganisms and heavy metal pollutants to reduce their toxicity [12]. However, due to the complex natural environment and differences in mining technology in the mining area, the metal content in various tailings is different [13]. To find more microbial strains that can be used for heavy metal remediation, it is necessary to investigate the microbial resources in different tail mining areas. Most research on heavy metals in soil focuses on how to remediate them, but neglects the emerging pollutant ARGs. Recent research shows that heavy metals will affect the relative abundance and distribution of ARGs in soil, and heavy metals may promote ARGs. Therefore, analyzing and supervising ARGs in tailings is of great significance [14].

One of the mega iron ore mines in China is the Jiangxi Xinyu-Ji'an iron ore mine located in the Xinyu region of Jiangxi Province [15], which is an important iron ore deposit in China. The mine has been in operation for many years, and the soil of the mine and the surrounding ecological environment is seriously contaminated with heavy metals. The study area of this paper was located in the Jiulong Mountain iron ore tailing area in Xinyu City, Jiangxi Province, where more than a dozen native tree species have been planted for the soil remediation of heavy metals. A preliminary survey found that Wetland pine (Pinus elliottii Engelmann), Chinese fir (*Cunninghamia lanceolata (Lamb.) Hook.*) and Alder (*Alnus cremastogyne* Burkill) had good growth performance in this mining area, but the remediation effect and mechanism on heavy metal pollution were not clear. Therefore, this study explored the remediation effect of heavy metals, soil microbial diversity, and community characteristics of the three tree species, and explored the characteristics of microbial resources in the iron tailing area before and after plant planting, to provide a theoretical basis and reference for the combined plant-microbial remediation in an iron tailing area.

## 2. Material and Methods

### 2.1. Study Area Overview

Jiulong Iron Tailing Area (27°38′25.37″ N, 114°50′27.06″ E) is located in Jiulong Towndisange, Xinyu City, Jiangxi Province, with a subtropical humid monsoon climate, with a year-round average temperature of 16.7 °C, a maximum temperature of 42 °C, and a minimum temperature of −7 °C; average annual precipitation of 1532 mm; and an elevation of 523.2 m. The tailings generated during ore collection, processing, and smelting have been piled up and exposed for a long time, resulting in rough soil texture and poor physical structure in the tailings area, making it difficult for vegetation to grow, and the ecological environment of the mine area has suffered serious damage.

The local forestry bureau has planted a variety of plants suitable for local growth in Jiulong Iron Tailing Area, but the effect is limited. The heavy metal contents in the plants and the rhizosphere and non-rhizosphere soils have been determined by ICP. Wetland Pine, Chinese fir, and Alder were selected as the research object of this study.

### 2.2. Sample Collection

In October 2019, root, stem, and leaf (stem and leaf parts are collectively referred to as above-ground parts in the following) samples and rhizosphere and non-rhizosphere soils samples of the three plants were collected. Each plant tissue and soil sample was sampled in three replicates. Three 10 × 10 m plots were set up in the growing area of each plant, five

sampling points were set up in each plot according to the diagonal sampling method, and 0–10 cm soil samples were collected and mixed. A total of 500 g were collected in soil bags according to the quartering method for soil chemical property analysis, and another 100 g were collected in 50 mL sterile centrifuge tubes for macrogenomic sequencing analysis. The samples were transferred to the laboratory through a portable refrigerator within 24 h. Soil samples for chemical analysis were brought back for immediate processing in the laboratory, and those for macrogenomics analysis were brought back and placed in an ultra-low temperature refrigerator ($-80$ °C) for subsequent measurements. Soil samples were collected as controls in the same way in open spaces without trees.

### 2.3. Chemical Property Determination

We removed roots, stones, and other debris from the soil samples, placed the treated soil samples in a ventilated place in the room, dried them naturally, and ground them with a 100-mesh sieve. The obtained soil samples were used as microwave digestion materials. The soil samples were microwave-digested using the triple acid method (6 mL concentrated $HNO_3$, 2 mL concentrated HCI, 1 mL HF), and the solution obtained from the digestion was transferred to a crucible. Then perchloric acid was added and placed on an electric hot plate at 180 °C to drive out the acid. The mixture in the crucible was evaporated to about 1 mL and the sample was transferred to a 50 mL centrifuge tube and the volume was fixed to 25 mL for determination. The contents of heavy metals Hg, Cr, Cu, Pb, and Zn were determined by inductively coupled plasma mass spectrometry (ICP-MS) and inductively coupled plasma optical emission spectroscopy (ICP-OES). The contents of heavy metal As were determined by atomic absorption spectrometry (ASS).

The plant samples were rinsed with sterile deionized water, placed in an oven at 105 °C for 0.5 h, dried at 80 °C for 72 h, ground through a 16-mesh sieve, and the resulting samples were used as microwave digestion materials. A total of 8 mL concentrated $HNO_3$ and 2 mL $H_2O_2$ were selected for microwave digestion of plant samples, and the method of driving acid and determination was the same as that of the soil samples.

### 2.4. DNA Extraction and Metagenomic Sequencing

Soil microbial metagenomics DNA were extracted from 0.2 g fresh soils using the Power Soil DNA Isolation Kit (MoBio Laboratories, Carlsbad, CA, USA), following the manufacturer's instructions. The DNA extraction of each sample was repeated five times and mixed to obtain enough DNA. After passing the genomic DNA test, the DNA was fragmented by mechanical interrupt method (ultrasonic), then the DNA fragments experienced purification, ends repair, adding A to 3′ ends, and adding adaptors. Next, agarose gel electrophoresis was used in fragment size selection and PCR amplification was used to construct the sequencing library. Once the constructed libraries passed the quality control, the qualified libraries were sequenced on the Illumina sequencing platform.

### 2.5. Data Processing and Analysis

#### 2.5.1. Soil Samples Chemical Property Analysis

Soil heavy metal contamination evaluation was performed using the risk screening values of the soil environmental quality-risk control standard for soil contamination of agricultural land (GB15618-2018) [16] as the evaluation criteria [17], using ICP-MS, ICP-OES and ASS to determine the concentrations of heavy metals Hg, Cr, Cu, Pb, Zn, and As. Then the results were compared with the risk screening values to determine the degree of heavy metal contamination in the collected soil. Excel 2013 and SPSS 22.0 were used for data analysis.

#### 2.5.2. Macrogenomic Sequencing Analysis

The raw reads obtained by sequencing contain low-quality sequences. To ensure the quality of information analysis, it is necessary to filter them to obtain clean reads for subsequent information analysis. Using the software Trimmomatic (http://www.

usadellab.org/cms/?page=trimmomatic accessed on 29 June 2023, Version 0.40), the raw sequences obtained from Illumina sequencing are quality controlled and filtered to obtain high-quality quality control data. Macrogenome assembly was performed using the software MEGAHIT (https://github.com/voutcn/megahit accessed on 29 June 2023) to filter contig sequences shorter than 300 bp [18]. The following various technical validations were performed to ensure data quality. Sequencing quality (including sequence quality per base, sequence content per base, and N content per base) was assessed by FAstQC 0.10 [19]. Genome assembly quality results were assessed using the software QUAST (http://bioinf.spbau.ru/quast accessed on 29 June 2023) [20]. MetaGeneMark software (http://exon.gatech.edu/meta_gmhmmp.cgi accessed on 29 June 2023, Version 3.26) was used to identify coding regions in the genome using default parameters [21]. Redundancy was removed using MMseqs2 software (https://github.com/soedinglab/mmseqs2 accessed on 29 June 2023, Version 12-113e3) with a similarity threshold set to 95% and coverage threshold set to 90%, and the gene with the longest length of each cluster was selected as the representative gene to construct a non-redundant gene set [22]. BLASTP (BLAST Version 2.2.28+, http://blast.ncbi.nlm.nih.gov/Blast.cgi accessed on 29 June 2023) was selected to compare the sequences of the gene sets in the obtained samples with the National Center for Nonredundant Biotechnology Information (NCBI-NR) database, and the results obtained were annotated for classification, and the comparison parameter e-value was set to $1 \times 10^{-5}$ [23]. The abundance of each taxonomic level was described as TPM and clustered using the Bray–Curtis method [24]. The relative abundance of microorganisms at each taxonomic level was calculated by analyzing the number of single genes annotated as the corresponding taxonomic level to obtain the microbial richness at different taxonomic levels. To reduce the influence of rare organisms, we considered only the top 10 microorganisms in terms of abundance and classified the rest as "other".

## 3. Results and Analysis

### 3.1. Effect of Three Tree Species Planting on the Heavy Metal Content of the Soil in Mining Areas

The average concentrations of As, Hg, Cr, Cu, Pb, and Zn in the rhizosphere and non-rhizosphere soil samples of the three plants were determined and the results were shown in Table 1. According to the Soil Environmental Quality Risk Control Standards for Soil Contamination on Agricultural Land (GB 15618-2018) [16], the comparison results of the two heavy metals in the samples showed that the concentrations of Cu and Zn in the samples were higher than the standard values, and the exceeding multiples were 1.12 and 1.14, respectively. The contents of As, Hg, Cu, and Pb in the non-rhizosphere soil of Chinese fir were significantly lower than those in the rhizosphere soil, while the contents of Cr and Zn were higher than those in the rhizosphere soil, indicating that Chinese fir had a strong remediation ability for As, Hg, Cu, and Pb, and a weaker remediation ability for Cr and Zn. The contents of As, Cr, Pb, and Zn in the non-rhizosphere soil of Wetland Pine were significantly lower than those in the rhizosphere soil, while the contents of Hg and Cu were higher than those in the rhizosphere soil, indicating that Wetland Pine had better restoration ability for As, Cr, Pb, and Zn, and a general restoration ability for Hg and Cu. On the other hand, Alder focused more on the restoration of Zn, reasoning that only the content of Zn was lower than that of rhizosphere soil in its non- rhizosphere soil.

**Table 1.** The concentration of heavy metals in the soil of Jiulong Mountain (Mean $\pm$ SE).

| Species | Position | As mg·kg$^{-1}$ | Hg μg·kg$^{-1}$ | Cr mg·kg$^{-1}$ | Cu mg·kg$^{-1}$ | Pb mg·kg$^{-1}$ | Zn mg·kg$^{-1}$ |
|---|---|---|---|---|---|---|---|
| Chinese fir | Rhizosphere | 2.05 ± 0.04 | 15.23 ± 0.26 | 64.84 ± 3.28 | 82.47 ± 2.45 | 50.35 ± 3.69 | 238.34 ± 4.37 |
| | Non-rhizosphere | 1.52 ± 0.02 | 12.64 ± 0.23 | 66.06 ± 3.77 | 79.62 ± 1.74 | 45.94 ± 1.71 | 246.06 ± 2.38 |
| Wetland Pine | Rhizosphere | 4.69 ± 0.08 | 15.35 ± 0.47 | 107.52 ± 1.81 | 25.81 ± 0.13 | 63.15 ± 2.88 | 282.68 ± 6.01 |
| | Non-rhizosphere | 4.32 ± 0.04 | 17.53 ± 0.04 | 104.27 ± 4.89 | 26.12 ± 0.98 | 60.00 ± 1.60 | 270.24 ± 8.68 |
| Alder | Rhizosphere | 1.63 ± 0.08 | 21.76 ± 0.25 | 118.89 ± 2.76 | 60.00 ± 2.04 | 65.53 ± 3.35 | 170.83 ± 41.78 |
| | Non-rhizosphere | 1.64 ± 0.09 | 24.62 ± 0.13 | 124.39 ± 4.61 | 63.44 ± 0.96 | 71.84 ± 2.74 | 164.96 ± 36.37 |

**Table 1.** *Cont.*

| Species | Position | As mg·kg$^{-1}$ | Hg μg·kg$^{-1}$ | Cr mg·kg$^{-1}$ | Cu mg·kg$^{-1}$ | Pb mg·kg$^{-1}$ | Zn mg·kg$^{-1}$ |
|---|---|---|---|---|---|---|---|
| Comparison | | 2.28 ± 0.09 | 37.34 ± 0.17 | 106.02 ± 2.08 | 26.80 ± 1.46 | 64.28 ± 1.07 | 114.14 ± 3.49 |
| Soil environmental quality: risk control standard for soil contamination of agricultural land (GB15618-2018) [16] | | 40 | 1300 | 150 | 50 | 70 | 200 |

### 3.2. Heavy Metal Content in Plants

The contents of As, Hg, Cr, Cu, Pb, and Zn in the three plants were determined and the results are shown in Table 2. The contents of each heavy metal in the above-ground parts of the plants ranged from As 107.08 to 248.65 μg·kg$^{-1}$, Hg 17.84 to 25.38 μg·kg$^{-1}$, Cr 14.81 to 23.37 mg·kg$^{-1}$, Cu 11.77 to 38.77 mg·kg$^{-1}$, Pb 50.58 to 82.74 mg·kg$^{-1}$, and Zn 96.01 to 183.51 mg·kg$^{-1}$. The contents of each heavy metal in the roots of the plants ranged from As 155.35 to 269.67 μg·kg$^{-1}$, Hg 5.34 to 9.77 μg·kg$^{-1}$, Cr 25.18 to 26.70 mg·kg$^{-1}$, Cu 6.45 to 21.07 mg·kg$^{-1}$, Pb 11.52 to 29.34 mg·kg$^{-1}$, and Zn 31.36 to 80.34 mg·kg$^{-1}$. The range of each heavy metal in the plant body was as follows: As 404.00–462.14 μg·kg$^{-1}$, Hg 23.14–35.15 μg·kg$^{-1}$, Cr 41.51–48.55 mg·kg$^{-1}$, Cu 19.65–59.84 mg·kg$^{-1}$, Pb 79.92–100.73 mg·kg$^{-1}$, and Zn 130.51–263.85 mg·kg$^{-1}$. The order of heavy metal contents in plants from high to low is Zn > Pb > Cr > Cu > As > Hg. The contents of As, Hg, Cr, Cu, Pb, and Zn in the three plants did not reach the critical content of hyperaccumulators [25], but Cr and Pb all exceeded the normal levels of heavy metal elements in plants. They showed high tolerance to Cr and Pb, all of which can be selected as remediation plants for Cr and Pb. The Cu and Zn contents in Alder exceeded the normal content, reflecting a higher tolerance to Cu and Zn, and could be selected as restoration plants for Cu and Zn. The As and Hg contents in the three plants were within the normal range because the As and Hg contents in the soil of the sample site were low.

**Table 2.** The contents of heavy metals in plants of Jiulongshan Mountain (Mean ± SE).

| Species | Component | As μg·kg$^{-1}$ | Hg μg·kg$^{-1}$ | Cr mg·kg$^{-1}$ | Cu mg·kg$^{-1}$ | Pb mg·kg$^{-1}$ | Zn mg·kg$^{-1}$ |
|---|---|---|---|---|---|---|---|
| Chinese fir | Aboveground | 248.65 ± 0.67 | 25.38 ± 0.40 | 23.37 ± 0.28 | 11.77 ± 0.07 | 82.74 ± 3.11 | 96.01 ± 0.73 |
| | Root | 155.35 ± 0.93 | 9.77 ± 0.05 | 25.18 ± 0.32 | 7.88 ± 0.11 | 17.99 ± 1.26 | 34.50 ± 0.38 |
| Wetland Pine | Aboveground | 193.80 ± 0.59 | 18.86 ± 0.36 | 15.93 ± 0.30 | 17.36 ± 0.18 | 80.65 ± 1.89 | 100.23 ± 0.89 |
| | Root | 268.34 ± 0.21 | 9.06 ± 0.08 | 26.19 ± 0.18 | 6.45 ± 0.03 | 11.52 ± 0.66 | 31.36 ± 0.10 |
| Alder | Aboveground | 107.08 ± 0.88 | 17.84 ± 0.23 | 14.81 ± 0.21 | 38.77 ± 0.37 | 50.58 ± 1.37 | 183.51 ± 1.92 |
| | Root | 269.67 ± 0.21 | 5.34 ± 0.26 | 26.70 ± 0.52 | 21.07 ± 0.36 | 29.34 ± 0.39 | 80.34 ± 0.51 |
| Normal content in plants | | <1000.00 | <100.00 | 0.20~3.00 | 0.40~45.8 | 0.10~41.70 | 1.00~160.00 |
| The critical value of hyperaccumulator | | 1,000,000.00 | 10,000.00 | 1000.00 | 1000.00 | 1000.00 | 10,000.00 |

### 3.3. Analysis of Soil Microbial Community Composition of the Three Plants

#### 3.3.1. Species Annotation and Statistics

To reveal the microbial community composition, information on the species composition and relative abundance of the samples was obtained from sequences compared to the Nr database based on macrogenomic non-redundant genes. The 6 samples contained 176 phyla, 149 classes, 324 orders, 740 families, 3072 genera, and 23,915 species in 7 realms.

#### 3.3.2. Species Diversity at the Phylum Level

The number of microbial species composition at the phylum level detected in the Jiulong Iron Tailings Area by database comparison analysis was 163 (J1, control), 169 (J2, Wetland Pine rhizosphere), 172 (J3, Wetland Pine non-rhizosphere), 173 (J4, Chinese fir rhizosphere), 168 (J5, Alder rhizosphere), and 171 (J6, Alder non-rhizosphere), for a total of 176 phyla. The top 10 microorganisms in relative abundance percentages in the 6 samples,

except for species that did not receive taxonomic annotation, are shown in Figure 1. The four dominant phyla in the six samples were Proteobacteria, Acidobacteria, Actinobacteria, and Chloroflexi. The top three dominant phyla (sum of relative abundance > 50%) were Proteobacteria, Acidobacteria, and Actinobacteria.

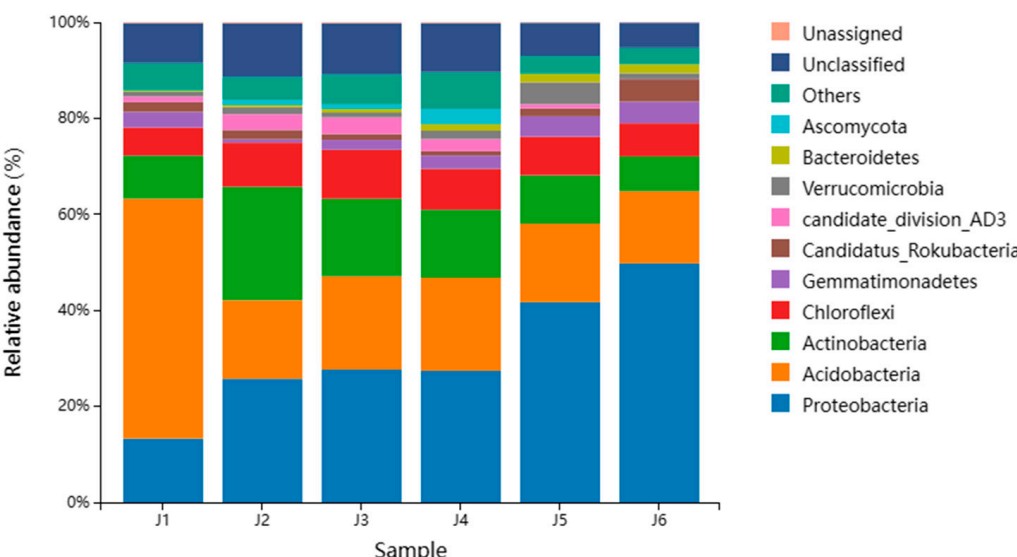

**Figure 1.** Histogram of relative abundance of species composition at the phylum level.

The phylum Acidobacteria can degrade plant residues, participate in iron cycling, and participate in physiological and biochemical reactions such as single-carbon compound metabolism [26]. The phylum Proteobacteria can participate in soil formation, activate and convert heavy metal forms, improve plant uptake of heavy metals, change the physicochemical properties of rhizosphere soil, and promote nutrient uptake by plants [27]. The phylum Actinobacteria can produce a variety of antibiotics (e.g., streptomycin, erythromycin, etc.), indoleacetic acid, and iron carriers with nitrogen fixation ability to inhibit plant pathogens and promote plant growth [28]. Although this general pattern of dominant phylum was observed in all six samples, the most abundant phylum varied from sample to sample. For example, for J1, the Acidobacteria phylum appears to be the most abundant phylum with a relative abundance percentage of 45.55%, which is absolutely dominant, followed by the phylum Proteobacteria with a relative abundance percentage of 12.11%. For J2, J3, J4, J5, and J6, the highest relative abundance percentages were found for the phylum Proteobacteria with 22.72%, 23.56%, 23.06%, 36.99%, and 44.56%, respectively, while the relative abundance percentages for the phylum Acidobacteria were 14.45%, 16.56%, 16.23%, 14.51%, and 13.49%, respectively. The relative abundance percentages of the six samples in the phylum Actinobacteria were 8.16% (J1), 20.85% (J2), 13.75% (J3), 11.92% (J4), 8.91% (J5), and 6.46% (J6).

J2 and J3, as the rhizosphere and non-rhizosphere soil samples of Wetland Pine, were consistent in the dominant phylum with the highest relative abundance, both being the phylum Proteobacteria, and the proportion is also relatively close—22.72% and 23.56%. In the species of dominant phylum, J2 is unique to Verrucomicrobia and J3 is unique to Gemmatimonadetes. Compared with the other five samples, the unique dominant phylum of J4 is Ascomycota. Both J5 and J6, as rhizosphere and non-rhizosphere soil samples of Alder, had eight dominant phyla, and the highest abundance of dominant phyla were both Proteobacteria. However, the percentages were somewhat different, being 36.99% and 44.56%, respectively.

To further confirm the differences in microbial community composition of the six samples at the phylum level, the Bray–Curtis algorithm was selected to perform species similarity analysis of the six samples at the gate level based on NMDS [29] (Figure 2). According to the NMDS plot, it can be seen that J1 is far away from the other five samples,

and there are obvious differences; J2 and J3, as the rhizosphere and non-rhizosphere soil samples of Wetland Pine, are closer in the plot, with more repeatable samples and higher species composition similarity; J3 and J4, as the non-rhizosphere soil of Wetland Pine, are closer to the rhizosphere soil of Chinese fir, with more repeatable samples and higher species composition similarity; J5 and J6, as the rhizosphere and non-rhizosphere soil samples of Alder, the distance between the two points was closer and more distant from the other four points, indicating that the rhizosphere and non-rhizosphere soil samples of Alder had higher species similarity and more significant differences from the other soil samples. In conclusion, the distance between J1 and the five samples of planted plants was far and there were significant differences; the distance between the rhizosphere soil samples of different species was far and there were significant differences, and the species similarity between the rhizosphere and non-rhizosphere soils of each plant (J2 and J3, J5 and J6) had a high degree of similarity in species composition.

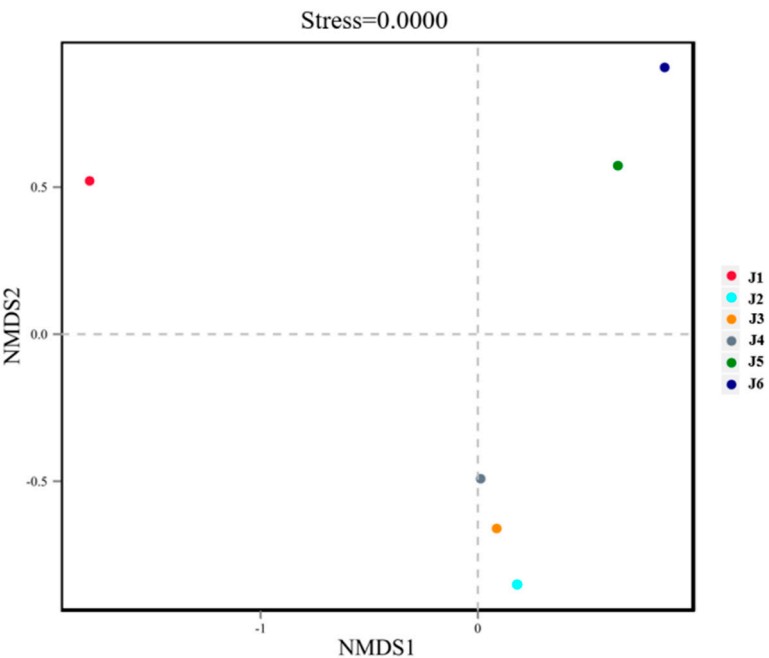

**Figure 2.** Phylum−level species NMDS analysis diagram. Note: The points in the figure represent each sample, respectively, and the distance between the points represents the degree of difference. The closer the distance on the coordinate graph, the higher the similarity.

### 3.3.3. Species Diversity at the Genus Level

The number of microbial species composition at the genus level detected in the iron tailing area of Jiulongshan by database comparison analysis was 2052 (J1), 2560 (J2), 2603 (J3), 2555 (J4), 2256 (J5), and 2445 (J6), for a total of 3072 genera. The top 10 microorganisms in relative abundance percentages in the 6 samples, except for species that did not receive taxonomic annotation, are shown in Figure 3. Taking the relative abundance >1% as the criterion, the six samples had a total of seven dominant genera, in the order of *Bradyrhizobium*, *Sphingomonas*, *Streptomyces*, *Ktedonobacter*, *Variovorax*, *Paraburkholderia*, and *Rhizobacter*. The endemic genera present only in that sample were detected in each sample, corresponding to 24 (J1), 36 (J2), 53 (J3), 50 (J4), 31 (J5), and 77 (J6), in that order, indicating that the cultivation of the three plants was effective in increasing the number of soil genera compared to the soil samples without plants (Figure 3).

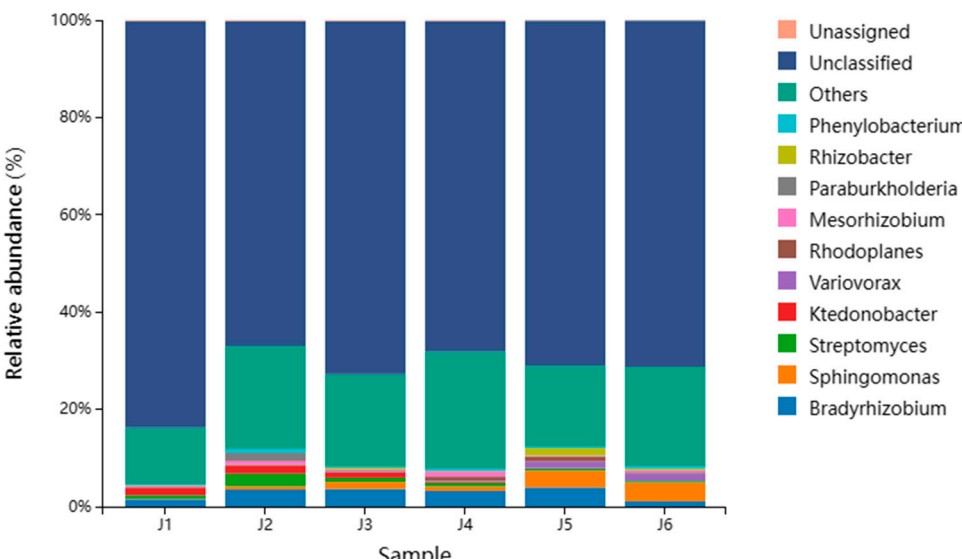

**Figure 3.** Genus-level species composition relative abundance histogram.

The dominant genera of the six samples varied in number and percentage and were not consistent, with six of the seven dominant genera coming from the phylum Proteobacteria and one from the phylum Actinobacteria. The genus *Bradyrhizobium* is from the phylum Proteobacteria and can bind to host plant roots to produce rhizobia with nitrogen fixation capacity [30]. *Sphingomonas* are from the phylum Proteobacteria and are involved in the soil nitrification–denitrification processes, which can decompose ammonia nitrogen to nitrite nitrogen and convert ammonia nitrogen to nitrogen gas [31]. *Streptomyces* are from the phylum Actinobacteria and can produce antibiotics such as streptomycin, tetracycline, and erythromycin [32]. *Ktedonobacter* are from the phylum Proteobacteria and can hydrolyze polysaccharides such as starch, cellulose, and xylan [33]. The genus *Paraburkholderia* is from the phylum Proteobacteria and has beneficial effects on host plants by producing phytohormones, antibiotics, and catabolic enzymes, fixing atmospheric nitrogen, dissolving soil mineral nutrients, and inducing systemic resistance in host plants [34]. *Variovorax* from the phylum Proteobacteria can play an important role in plant stress resistance by regulating growth factor concentration to ensure plant root health [35]. *Rhizobacter* is from the phylum Aspergillus and can degrade organic pollutants and remediate heavy metals and can also symbiotically nodulate and fix nitrogen with legumes to promote plant growth and enhance plant resilience [36].

J1 had two dominant genera, *Bradyrhizobium* (1.38%) and *Ktedonobacter* (1.43%); J2 had four dominant genera, *Bradyrhizobium* (3.09%), *Streptomyces* (2.31%), *Ktedonobacter* (1.48%), and *Paraburkholderia* (1.48%), of which *Bradyrhizobium* had a significant advantage with more than 3%; J3 had two dominant genera, *Bradyrhizobium* (3.10%) and *Sphingomonas* (1.22%), of which *Bradyrhizobium* accounted for more than 3% and had a clear advantage. J4 had only one dominant genus of *Bradyrhizobium* (2.72%). J5 has four dominant genera: *Bradyrhizobium* (3.49%), *Sphingomonas* (3.09%), *Variovorax* (1.22%) and *Rhizobacter* (1.35%), of which *Bradyrhizobium* and *Sphingomonas* are more than 3%. J6 has two dominant genera: *Sphingomonas* (3.47%) and *Variovorax* (1.35%), of which *Sphingomonas* accounts for more than 3%.

J1 was significantly different from the other five samples in terms of the number and types of dominant genera, and the dominant genera did not exceed 3% in their samples. As the rhizosphere and non-rhizosphere soil samples of Wetland Pine, J2 and J3 have significant differences in the number of dominant genera, while they have commonness in the bacterial genus with the highest proportion of their samples (more than 3%, the same below), both of which are *Bradyrhizobium*. J5 and J6, as rhizosphere and non-rhizosphere soil samples of Alder, have obvious differences in the number of dominant genera. J5 and J6, as rhizosphere and non-rhizosphere soil samples of Alder, had the same dominant genus

with the highest proportion of their samples, J5 being *Bradyrhizobium* and *Sphingomonas* and J6 being *Sphingomonas*. The bacteria with the highest proportion of their samples are *Bradyrhizobium* and *Sphingomonas* (Figure 4).

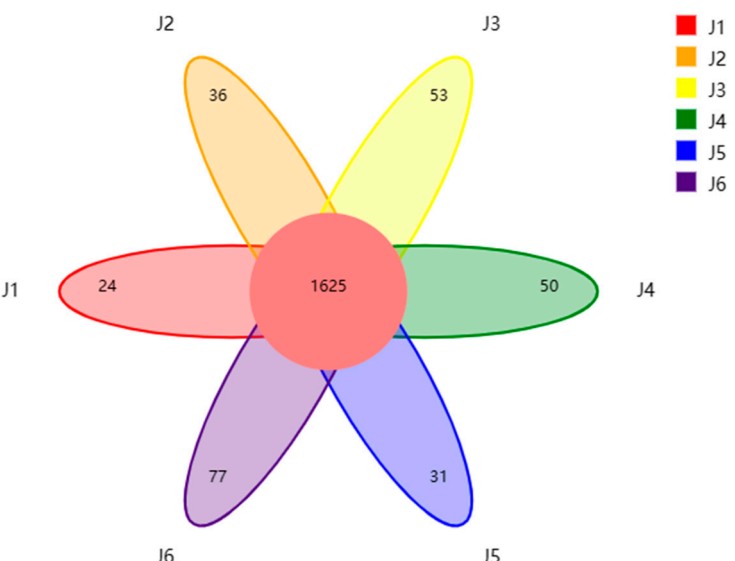

**Figure 4.** Venn diagram of the genus-level species composition number.

### 3.4. Soil Antibiotic Resistance Genes Characteristics

The antibiotic resistance genes (ARGs) in microorganisms can respond to the inducement of heavy metals. Through CARD database comparison analysis, it was detected that the number of antibiotic resistance genes (ARGs) in the Jiulong iron tailings tail mining area was 286 (J1), 371 (J2), 333 (J3), 311 (J4), 293 (J5), and 405 (J6) in the control soil, rhizosphere, and non-rhizosphere soils of the three tree species, respectively. The number of antibiotic resistance gene types in the six samples is 27 (J1), 25 (J2), 28 (J3), 24 (J4), 22 (J5), and 24 (J6), in sequence.

It can be seen from Figure 5 that in the analysis of drug categories, ARGS in six samples showed resistance to nine different drugs, including fluoroquinolones, amino Coumarin, Carbapenem, Chloramphenicol, Macrolide, peptides, glycopeptides, tetracyclines and Aminoglycosides. Among the six samples, the relative abundance of drug resistance genes with the highest relative abundance is aminoglycosides (J1, J2, J3, J4, J5) and Peptide (J6). From the perspective of mechanism, antibiotic efflux is the most important type in tailings.

In order to further explore the differences in resistance gene characteristics among the six samples, the Bray–Curtis algorithm was selected on NMDS for functional similarity analysis, with a stress value of 0.0000 (<0.2), proving that NMDS has a good overall dimensionality reduction effect, and the final results are available (Figure 6). The distance between J2 and J3 as rhizosphere and non-rhizosphere soil samples of Wetland pine is relatively far, indicating a low functional similarity between them. The rhizosphere and non-rhizosphere soil samples of Alder are closely spaced, indicating that their functional similarity is high, but there are still some differences. In summary, among the rhizosphere and non-rhizosphere soil samples of the same tree species, there are significant differences between Wetland pine, with lower functional similarity, while Alder has smaller differences and higher functional similarity. The rhizosphere soil samples of different tree species have significant differences and low functional similarity.

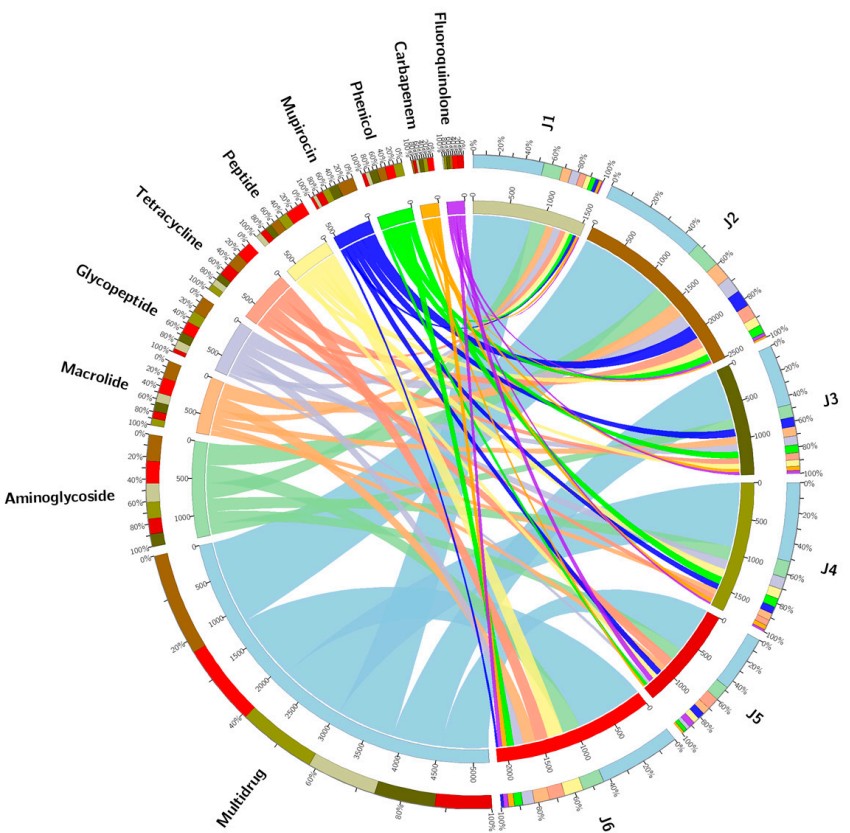

**Figure 5.** Loop map of antibiotic resistance genome.

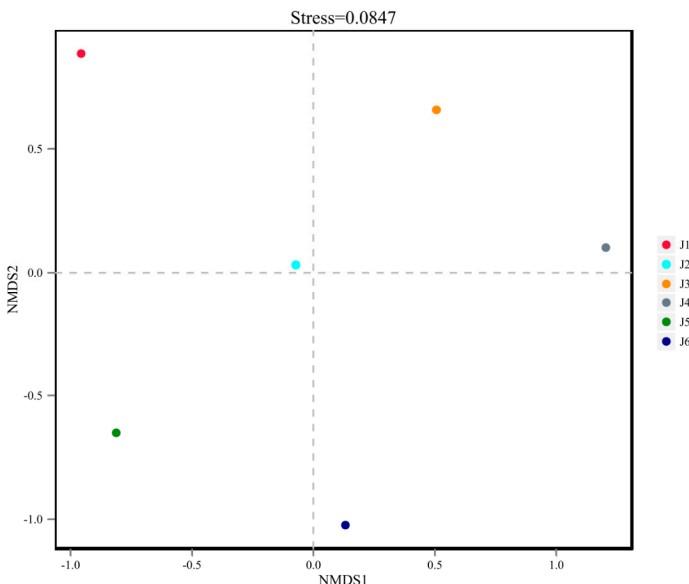

**Figure 6.** NMDS analysis diagram of antibiotic resistance genes.

## 4. Discussion

### 4.1. Heavy Metal Enrichment and Transfer Capacity of Plants

The three remediation plants selected in this study showed different degrees of resistance to heavy metal compound pollution in the tailings area habitat, and all of them can be selected as remediation plants for heavy metal pollution in the iron tailings area. All three plants showed high tolerance to Cr and Pb, strong transfer ability to Hg, Cu, Pb, and Zn, and weak transfer ability to Cr, while the enrichment ability was focused on its emphasis. Chinese fir and Wetland Pine had better enrichment ability for Hg and Pb, and Alder had

better enrichment ability for Zn. Some studies found that the root tissues of Wetland Pine have a certain sorption ability for Cr, Cu, Zn, and other heavy metals [37], which is the same as our findings. In this study, we found that Wetland Pine has a weak uptake capacity for Cu, which is the same as the results of existing studies. Some scholars have found that a certain heavy metal element in plants is proportional to the heavy metal element in soil [38]. In the present study, the characteristics of the heavy metal content in the three plants were not the same as the soil heavy metal content, indicating that the enrichment of heavy metals by plants is not only related to the heavy metal content in the soil but also related to many factors such as plant characteristics, physicochemical properties, soil physicochemical properties, concentration, morphology, and toxicity of heavy metals in the soil, thus leading to large differences in the results of many studies.

In the case of widespread soil contamination, plants with small limitations in metal transport from root to stem are more suitable for phytostabilized recovery and tailings soil remediation [39]. However, the extremely fluctuating environment of the tailings area (e.g., heavy metal toxicity, poor fertility, poor water holding capacity, and high acidity) may hinder the process of ecological recovery by direct planting of vegetation [40], and the ability of trees and shrubs to grow stably is a key factor in the recovery of mine vegetation. Most of the super-enriched planets found so far are herbaceous plants, so to better remediate heavy metal pollution in this area [41], herbaceous plants suitable for local growth can be selected for planting in further studies [42]. By enriching the species diversity and forming a more stable ecosystem, one improve the efficiency of heavy metal remediation and reduce soil erosion in tailings ponds, and soil conditioners can also be combined to alleviate the inhibition of plant growth by heavy metal stress [43].

### 4.2. Microbial Community Composition at Different Levels

4.2.1. Microbial Community Composition at the Phylum Level

At the phylum level, the species and abundance percentage of the dominant phylum in J1, a soil sample without planting, were significantly different from those of other planted soil samples, indicating that the planting of three plants had a significant effect on the soil microbial phylum composition. The phylum with absolute dominance in J1 was Acidobacteria (45.55%), followed by Proteobacteria (12.11%). For J2, J3, J4, J5, and J6, the highest relative abundance percentages were found for the phylum Proteobacteria with 22.72%, 23.56%, 23.06%, 36.99%, and 44.56%, respectively, and for the phylum Acidobacteria with 14.45%, 16.56%, 16.23%, 14.51%, and 13.49%, respectively. The difference in the relative abundance of Acidobacteria phylum between J1 and the remaining five samples ranged from 28.99% to 32.06%, and Proteobacteria phylum ranged from 10.61% to 32.45%. This shows that plant cultivation significantly promotes the increase of the relative abundance of Proteobacteria phylum in the soil. The phylum Proteobacteria promotes the uptake of heavy metals by plants, indicating that Amoebacteria phylum is a key microbial phylum for the remediation of heavy metal pollution in the study area.

The relative abundance percentages of the phylum Actinobacteria in the six samples were in the following order: 8.16% (J1), 20.85% (J2), 13.75% (J3), 11.92% (J4), 8.91% (J5), and 6.46% (J6). Compared to the other samples, antimicrobial production seemed to be more active in the rhizosphere soil of Wetland Pine, presumably possibly containing more antimicrobial substances. The reason is that the relative abundance of Actinobacteria in J2 reached more than 20%, which is significantly higher than that of Acidobacteria (14%). In the other four samples, the Actinomycetes phylum was only about 6%–13%, which is much lower than the relative abundance percentage value of the second place.

4.2.2. Microbial Community Composition at the Genus Level

At the genus level, the dominant genera of the six samples are different in number and proportion. In addition to the dominant genera, each sample can detect multiple endemic genera only owned by itself. It shows that after planting different dominant plants, the rhizosphere and non-rhizosphere soils of corresponding plants will produce

different dominant bacteria and unique microbial genera. Therefore, it is speculated that the ways of remediation of heavy metals by different plants are different. The six samples contain seven dominant genera, including six genera from Proteobacteria and one genus from Actinobacteria. It shows that the planting of plants promotes the increase of the relative abundance of Proteobacteria in soil. Proteobacteria is the key community for the remediation of heavy metal contaminated soil. This is consistent with the results of microbial community composition analysis at the phylum level.

J1 was significantly different from the other samples in terms of abundance and species of dominant genera. The dominant bacteria were *Bradyrhizobium* (1.38%) and *Ktedonobacter* (1.43%), which accounted for a similar proportion. There were no obvious dominant bacteria, while the other samples had obvious dominant bacteria. Rhizosphere and non-rhizosphere soils of the same plant have commonalities in the genus with the highest relative abundance. Wetland Pine belongs to the genus *Bradyrhizobium*, which are 3.09% (J2) and 3.10% (J3). The dominant genus of Alder is *Sphingomonas*, which are 3.09% (J5) and 3.47% (J6). The dominant genus with the highest proportion of rhizosphere soil of different species of plants is the same, which is *Bradyrhizobium*. It shows that the survival of the three plants in heavy metal polluted habitats is similar to the way of dealing with heavy metals. The number of dominant genera of J2, J3, J4, J5, and J6 is 4, 2, 1, 4, and 2, which indicates that different plant species will produce differences in the number of dominant genera. The number of dominant genera in rhizosphere soil of the same plant is significantly higher than that in non-rhizosphere soil, indicating that the microbial community in rhizosphere soil is more abundant in composition and function.

In terms of function, the dominant genus of J1 is mainly related to root nodule formation and nitrogen fixation. Compared with J1, J2 has two additional dominant genera, *Streptomyces* and *Paraburkholderia*, which can produce more plant hormones, antibiotics, and other substances, and have more beneficial effects on the host. It can be inferred that Wetland Pine can form a symbiotic relationship with rhizosphere soil microorganisms and promote each other. Compared with J1, J5 has three additional dominant genera: *Streptomyces*, *Variovorax*, and *Rhizobacter*. They can produce highly active iron carriers, have a stronger ability to hydrolyze organic and inorganic phosphorus, and promote plant growth through symbiotic nitrogen fixation. It is speculated that alder can form a symbiotic relationship with the rhizosphere soil microbial genera. However, microorganisms are easily restricted by different environmental factors (such as temperature, pH, oxygen content, etc.), which makes them have great differences in species diversity, community composition, and function. Considering the complexity of the soil ecosystem itself and the uncertain factors of the natural environment, this needs to be further determined.

### 4.3. Soil Antibiotic Resistance Genes Characteristics

Most research on heavy metals in soil focuses on how to remediate them, but neglects the emerging pollutant ARGs. Zhang Fengli's research shows that heavy metals will affect the relative abundance and distribution of ARGs in soil, and heavy metals may promote ARGs [14]. Therefore, analyzing and supervising ARGs in tailings is of great significance.

There are significant differences in the quantity and category of ARGs among the six samples, and there are significant differences in the quantity of ARGs among the rhizosphere and non-rhizosphere soil samples of the three tree species. This indicates that the planting of the three tree species significantly affects the quantity and category of ARGs in the soil, and the planting of different tree species will have different effects on the quantity and category of ARGs in the soil. Among the six samples, the relative abundance of drug resistance genes with the highest relative abundance is aminoglycosides (J1, J2, J3, J4, J5) and Peptide (J6), indicating that the planting of three tree species has significantly caused changes in the function of resistance genes in the soil. As the rhizosphere and non-rhizosphere soil samples of Wetland pine, J2 and J3 have different resistance genes with the highest relative abundance, while J5 and J6 are the rhizosphere and non-rhizosphere soil samples of Alder, respectively. The resistance genes with the highest relative abundance

have similarities in species, but are somewhat different from J2 and J3, indicating that planting of different tree species has different effects on the characteristics of resistance genes in soil. The results of NMDS functional similarity analysis are also consistent with the above conclusions. From the perspective of drug resistance, ARGs has been observed to be resistant to a variety of antibiotics commonly used in clinical practice (such as tetracyclines, Macrolide, etc.). At the same time, microorganisms carrying relevant resistance genes can continuously migrate in the soil and spread to the surrounding environment, thereby increasing their harm, which is an issue that needs attention.

## 5. Conclusions

Wetland Pine and Chinese fir can be used as the pioneer species for Hg and Pb remediation, and Alder can be used as the remediation species and intercropping for Zn. Based on the analysis of the level of phylum and genus, the dominant microbial phylum in the rhizosphere and non-rhizosphere soil of the three tree species is Proteobacteria. The planting of the three tree species has promoted the increase of the relative abundance of Proteobacteria phylum microorganisms in the soil. Proteobacteria phylum can promote plant growth and absorb heavy metals, and the interaction between the two ultimately achieves a better effect of heavy metal absorption by plants. There are significant differences in the number and type of antibiotic resistance genes in the rhizosphere and non-rhizosphere soils of three typical heavy metal remediation tree species before and after planting. The planting of different tree species will have different effects on the characteristics of antibiotic resistance genes in the soil.

**Author Contributions:** Conceptualization, L.H., L.S. and Q.S.; Methodology, L.H., W.W., L.S., Q.W. and X.L.; Software, L.H.; Validation, L.H. and Q.W.; Formal analysis, L.H.; Investigation, L.H., W.W., L.S., X.L., Y.Z. and Q.S.; Data curation, L.H., W.W., Q.W., X.L. and Y.Z.; Writing—original draft, W.W.; Writing—review & editing, L.H.; Visualization, L.H.; Supervision, L.H. and Q.S.; Project administration, L.H. and Q.S.; Funding acquisition, Q.S. All authors have read and agreed to the published version of the manuscript.

**Funding:** This research was funded by the National Key Research and Development Program of China (No. 2022YFF130300503). And The APC was funded by the National Key Research and Development Program of China (No. 2022YFF130300503).

**Data Availability Statement:** Not applicable.

**Conflicts of Interest:** The authors declare no conflict of interest.

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
