# Peer review of "Phytoremediation Effect and Soil Microbial Community Characteristics of Jiulong Iron Tailings Area, Jiangxi"

_forests, doi:10.3390/f14091849_

Round 1

Reviewer 1 Report

Thank you for giving me the opportunity to revise the MS entitled “Phytoremediation effect and soil microbial community characteristics of Jiulong Mountain Iron Ore Tailing area in Xinyu and Jiangxi” by Hou and his/her colleagues that was submitted to “Forest”.  The MS submitted is suitable for Forest, and some interesting results were showed. However, there are several requirements that have to consider by the authors. In this regard, the following comments are requested to be addressed by the authors:

1.         The Abstract section needs to be carefully revised and refined. The innovation of the manuscript must be clearly stated in the text.

2.         Keywords needs to be carefully revised, such as “Heavy metal contamination; Iron ore tailing”

3.         Introduction needs to be carefully revised.

4.         Suggest carefully checking the format and language of the entire text.

5.         Line 85-103 is too verbose and not concise enough. And the key information was not provided.

6.         Line 186-196 What is the significance of studying this area, as most of the heavy metal content has met the standards?

7.        Please check the citation format of the main text and the reference format at the end of the text.

8.        I would suggest that the authors review and include the following recent studies to improve the manuscript.

Organic–inorganic composite modifiers enhance restoration potential of Nerium oleander L. to lead–zinc tailing: application of phytoremediation. Environ Sci Pollut R. 2023, 30, 56569–56579.

Lead responses and tolerance mechanisms of Koelreuteria paniculata: A newly potential plant for sustainable phytoremediation of Pb-contaminated soil [J]. International Journal of Environmental Research and Public Health 2022, 19(22), 14968. 

Best regards,

Author Response

  1. The Abstract section needs to be carefully revised and refined. The innovation of the manuscript must be clearly stated in the text.

Response: The reviewer’s concen is valid. We have redrawn the Abstract in the revised manuscript.

  1. Keywords needs to be carefully revised, such as “Heavy metal contamination; Iron ore tailing”

Response: The authors would like to thank the reviewer for his/her advice. We have checked keywords in the revised manuscript.

  1. Introduction needs to be carefully revised.
    Response: The reviewer’s concen is valid. We have redrawn the Introduction in the revised manuscript.
  2. Suggest carefully checking the format and language of the entire text.

Response: The authors would like to thank the reviewer for his/her advice. We have checked the entire text in the revised manuscript.

  1. Line 85-103 is too verbose and not concise enough. And the key information was not provided.

Response: We have shortened these parts in the revised manuscript and provided the key information.

  1. Line 186-196 What is the significance of studying this area, as most of the heavy metal content has met the standards?

Response: The authors would like to thank the reviewer for his/her advice.The significance of studying this area has two reasons:(1) One of the mega iron ore mines in China is the Jiangxi Xinyu-Ji’an iron ore mine located in the Xinyu region of Jiangxi Province, which is an important iron ore deposit in China. (2)Cu and Zn in the samples were higher than the standard values, Pb and Cr in the samples were closed to the standard values. It is very meaningful to achieve good remediation effect of heavy metal pollution in tailings areas through the plant microbial combined remediation, and research cannot completely use the kinds of excessive heavy metals as the evaluation standard

  1. Please check the citation format of the main text and the reference format at the end of the text.

Response: We have checked the citation format of the main text and the reference format at the end of the text.

  1. I would suggest that the authors review and include the following recent studies to improve the manuscript.

Organic–inorganic composite modifiers enhance restoration potential of Nerium oleander L. to lead–zinc tailing: application of phytoremediation. Environ Sci Pollut R. 2023, 30, 56569–56579.

   Lead responses and tolerance mechanisms of Koelreuteria paniculata: A newly potential plant for sustainable phytoremediation of Pb-contaminated soil [J]. International Journal of Environmental Research and Public Health 2022, 19(22), 14968. 

Response: The authors would like to thank the reviewer for his/her advice. We added these studies in Introduction.

Reviewer 2 Report

1.      The abbreviated words J1 to J7 needs to be defined in abstract section. The other words like ARGS are also not defined.

2.      The abstract is more descriptive rather than presenting the results in numerical forms.

3.      Reduce the size of abstract.

4.      Remove the words mentioned in manuscript title from keywords list

5.      Introduction lacks enough description about phytoremediation. As a suggestion please read the following: Phytoremediation of toxic heavy metals in polluted soils and water of Dargai District Malakand Khyber Pakhtunkhwa, Pakistan. Brazilian Journal of Biology. 2024 • https://doi.org/10.1590/1519-6984.265278.

Removal of Iron(II) from Effluents of Steel Mills Using Chemically Modified Pteris vittata Plant Leaves Utilizing the Idea of Phytoremediation. Water 2022, 14, 2004. https://doi.org/10.3390/w14132004

6.      Please provide a rational of the study undertaken.

7.      Section 2.3 is written in present tense, please convert them to passive sentences.

8.      Section 2.4 a sentence never starts with a numerical digit please put approximate or about before it. Also there should be a space between the numerical value and its unit with the exception of temp and percent

9.      What far DNA extraction and Metagenome sequencing analysis has been performed. In my opinion they have nothing to do with phytoremediation. If cytotoxicity has performed then ok otherwise no need to include.

10.   Bioaccumulation and translocation factors have not been provided.

11.   Conclusion needs to be revised.

12.   Refences style is not uniform

ok. Minor correction needed 

Author Response

  1. The abbreviated words J1 to J7 needs to be defined in abstract section. The other words like ARGS are also not defined.

Response: As suggested by the reviewer, we defined the abbreviated words J1 to J7 and ARGs in the revised manuscript.

  1. The abstract is more descriptive rather than presenting the results in numerical forms.

Response: The reviewer’s concen is valid. We have redrawn the Introduction in the revised manuscript.

  1. Reduce the size of abstract.

Response: The authors would like to thank the reviewer for his/her advice.We have shortened these parts in the revised manuscript.

  1. Remove the words mentioned in manuscript title from keywords list

Response: The authors would like to thank the reviewer for his/her advice. We have rewritten keywords list in the revised manuscript.

  1. Introduction lacks enough description about phytoremediation. As a suggestion please read the following: Phytoremediation of toxic heavy metals in polluted soils and water of Dargai District Malakand Khyber Pakhtunkhwa, Pakistan. Brazilian Journal of Biology. 2024 • https://doi.org/10.1590/1519-6984.265278.

Removal of Iron(II) from Effluents of Steel Mills Using Chemically Modified Pteris vittata Plant Leaves Utilizing the Idea of Phytoremediation. Water 2022, 14, 2004. https://doi.org/10.3390/w14132004

Response: The authors would like to thank the reviewer for his/her advice. We added these studies in Introduction.

  1. Please provide a rational of the study undertaken.

Response: The reviewer’s concen is valid. The heavy metal pollution in tailings areas is a very noteworthy issue, and the combination of plant microbial combined remediation is an environmentally friendly remediation method. Therefore, it is necessary to study suitable plants and dominant microbial community.

  1. Section 2.3 is written in present tense, please convert them to passive sentences.

Response: The authors would like to thank the reviewer for his/her advice. We have rewritten Section 2.3 in the revised manuscript.

  1. Section 2.4 a sentence never starts with a numerical digit please put approximate or about before it. Also there should be a space between the numerical value and its unit with the exception of temp and percen.

Response: The authors would like to thank the reviewer for his/her advice. We have rewritten this part in the revised manuscript.

  1. What far DNA extraction and Metagenome sequencing analysis has been performed. In my opinion they have nothing to do with phytoremediation. If cytotoxicity has performed then ok otherwise no need to include.

Response: The authors would like to thank the reviewer for his/her advice. This study mainly focused on the role of plant microbial combined remediation of heavy metal pollution. Macrogenomic sequencing was necessary to identify the key microbial community involved in the plant microbial combined remediation.

  1. Bioaccumulation and translocation factors have not been provided.

Response: The authors would like to thank the reviewer for his/her advice. This was my fault, this study was not focused on bioaccumulation and translocation factors, I was delated the sentence in Abstract.

  1. Conclusion needs to be revised.

Response: The authors would like to thank the reviewer for his/her advice. We have rewritten this part in the revised manuscript.

  1. Refences style is not uniform.

Response: We have checked the citation format of the main text and the reference format at the end of the text.

Round 2

Reviewer 1 Report

ok to accept.

Author Response

Thank you very much for reviewing the manuscript, which has been very helpful in improving it.

Reviewer 2 Report

ok

Author Response

(The authors gave the same response as above.)
